# Meta-analysis shows positive effects of plant diversity on microbial biomass and respiration

Chen Chen[1], Han Y.H. Chen (ID) [1,2], Xinli Chen[1] & Zhiqun Huang (ID) [2,3]

Soil microorganisms are key to biological diversity and many ecosystem processes in terrestrial ecosystems. Despite the current alarming loss of plant diversity, it is unclear how plant species diversity affects soil microorganisms. By conducting a global meta-analysis with paired observations of plant mixtures and monocultures from 106 studies, we show that microbial biomass, bacterial biomass, fungal biomass, fungi:bacteria ratio, and microbial respiration increase, while Gram-positive to Gram-negative bacteria ratio decrease in response to plant mixtures. The increases in microbial biomass and respiration are more pronounced in older and more diverse mixtures. The effects of plant mixtures on all microbial attributes are consistent across ecosystem types including natural forests, planted forests, planted grasslands, croplands, and planted containers. Our study underlines strong relationships between plant diversity and soil microorganisms across global terrestrial ecosystems and suggests the importance of plant diversity in maintaining belowground ecosystem functioning.

---

[1] Faculty of Natural Resources Management, Lakehead University, 955 Oliver Road, Thunder Bay, Ontario P7B 5E1, Canada. [2] Key Laboratory for Humid Subtropical Eco-geographical Processes of the Ministry of Education, Fujian Normal University, Fuzhou 350007, China. [3] Institute of Geography, Fujian Normal University, Fuzhou 350007, China. Correspondence and requests for materials should be addressed to H.Y.H.C. (email: hchen1@lakeheadu.ca) or to Z.H. (email: zhiqunhuang@fjnu.edu.cn)

Soil microorganisms make up a significant portion of bio-diversity and play a critical role in many ecological processes, such as the cycling of carbon and nutrients[1–4]. Plant species diversity continues to decline due to anthropologically initiated environmental degradation[5], and the decrease in plant species diversity has been recognized as a major threat to eco-system functions and services[6,7]. Despite the critical influence of plant diversity on plant resources available for soil microorganisms and alterations to the micro-environment[8–13], the global effect of reduced plant diversity on soil microbial abundance, fungi:bacteria ratio, Gram-positive (G+) to Gram-negative (G−) bacteria ratio and function remains uncertain.

Plant diversity is an important factor for microbial biomass, fungi:bacteria ratio and G+:G− bacteria ratio. Soil microbial biomass can increase with plant diversity due to a greater amount of carbon and nutrient resources available for soil microorganisms from increased aboveground litterfall and belowground fine root mortality in species-rich plant communities[14–19]. Also, mixing functionally different grassland species leads to diverse root exudates, which enhance microbial diversity and biomass[20]. Further, plant diversity may increase the soil fungi:bacteria ratio as plant diversity was reported to increase fungal abundance, but not bacterial abundance, in a grassland experiment[21]. Moreover, plant species diversity increases soil moisture by higher leaf area index[22]. As G+ bacteria are more dominant than G− bacteria in dry conditions due to their thicker cell walls and capacity to form spores[23,24], the reduced soil moisture associated with lower plant diversity may increase G+:G− bacteria ratio. Therefore, our first hypothesis is that plant diversity would increase the total microbial biomass, bacterial and fungal biomass, and fungi:bacteria ratio, but decrease G+:G− bacteria ratio due to plant diversity induced increases of carbon inputs and soil moisture[14,17,18,22].

Plant diversity may affect soil microbial carbon to nitrogen (C:N) ratio and respiration. Soil microbial C:N ratio may increase with plant diversity via increased fungi:bacteria ratio[22] because fungi tend to have a higher C:N ratio than bacteria[25,26]. Additionally, microbial biomass and respiration tend to respond similarly to environmental changes[27,28], as well as to the variation of plant species diversity[29]. In the meantime, plant diversity may promote microbial metabolic efficiency and decrease microbial metabolic quotient (respiration-to-biomass ratio) by supplying diverse resources for microorganisms and modifying the micro-environment[16,30]. Therefore, our second hypothesis is that plant diversity would increase microbial C:N ratio and respiration but decrease metabolic quotient because of the effects of plant diversity on the relative abundance of fungi and bacteria[22], resources available for microorganisms, and the micro-environment[16,30].

The effects of plant diversity on above- and belowground productivity become progressively stronger over time[19,31]. This temporal change of diversity effects has been attributed to increasing interspecific complementarity and decreasing functional redundancy over time[32,33]. For microorganisms, there is a time lag in their response to changes in plant communities due to the accumulation of dead plant materials needed before the response of soil microorganisms[34,35]. The results of a long-term grassland experiment revealed that the effect of plant diversity on soil microorganisms was not significant until several years following plot establishment[29,35]. Thus, the divergence of plant diversity effects on microorganisms might have resulted from differences in time/stand ages, and our third hypothesis is that the increases of microbial biomass and respiration with plant diversity would be more pronounced at older stand age due to the time needed for the accumulation of dead plant materials[34,35].

Plant diversity effects on soil microorganisms may change with ecosystem types and the environment. For example, increases in plant productivity with plant diversity are stronger in nature than documented in experiments[36]. Nevertheless, whether the effects of plant diversity on soil microorganisms differ with ecosystem types varying from planted containers, croplands, planted grasslands, planted forests, and natural forests remain unexplored. Moreover, although a global meta-analysis has shown that the positive effects of tree species diversity and trait heterogeneity on aboveground productivity are consistent across the global forests[37], regional studies indicate that the positive effects of tree diversity tend to be stronger in colder climates[38,39]. A higher increase in plant productivity with diversity in colder climates may lead to higher amounts of dead plant materials for soil microorganisms and thus stronger increases of soil microbial biomass and respiration with plant diversity.

A previous synthesis of twelve studies in planted grasslands has revealed that soil microbial biomass increases significantly with plant species diversity, particularly in long-term experiments[40]. A previous meta-analysis in croplands found that the addition of one or more crop species to monocultures substantially increased soil microbial biomass[41]. It remains unclear, however, (1) whether the plant diversity effects on soil microbial biomass differ with ecosystem types, and spatially across global terrestrial ecosystems, and (2) how plant diversity influences fungi:bacteria ratio, G+:G− bacteria ratio, microbial C:N ratio, and respiration. We conducted a meta-analysis of 1332 paired observations of plant monocultures and mixtures from 106 studies to investigate the effects of plant diversity on total microbial, bacterial, and fungal biomass, fungi:bacteria ratio, G+:G− bacteria ratio, microbial C:N ratio, respiration, and metabolic quotient across global terrestrial biomes (Fig. 1; Supplementary Table 1, refs. [13,16,20–22,29,35,42–140]). We expected that: (1) plant mixtures would, on average, increase soil total microbial, bacterial and fungal biomass, fungi:bacteria ratio, microbial C:N ratio, and microbial respiration, but decrease G+:G− bacteria ratio and metabolic quotient; (2) the effect of plant mixtures would increase with the species richness in mixtures and stand age; (3) the effect of plant mixtures would differ with ecosystem types and change with climate. We show that plant mixtures increase soil total microbial, bacterial and fungal biomass, fungal:bacteria ratio but decrease G+:G− bacteria ratio, while having no effects on microbial C:N ratio and metabolic quotient. Moreover, microbial biomass and respiration increase more in mixtures with higher species richness and in older mixtures. These effects of plant mixtures are consistent across ecosystem types and global climates except a higher increase in microbial respiration in colder climates.

## Results

**The average effects of plant mixtures on microbial attributes.** Across all ecosystem types, total soil microbial biomass increased significantly on average by 12.5% (95% confidence interval, 7.9–17.1%; $P < 0.001$, Supplementary Table 2), bacterial biomass by 5.2% (0.2–10.1%; $P = 0.054$, Supplementary Table 2), and fungal biomass by 10.9% (4.0–17.8%; $P = 0.004$, Supplementary Table 2) in plant mixtures compared to the mean of constituent monocultures (Fig. 2). Plant mixtures increased fungi:bacteria ratio ($P = 0.036$, Supplementary Table 2). Also, plant mixtures, on average, increased soil microbial respiration by 13.2% (8.1–18.2%; $P < 0.001$, Supplementary Table 2). G+:G- bacteria ratio, microbial C:N ratio, and metabolic quotient showed no significant responses to plant mixtures (Fig. 2; Supplementary Table 2).

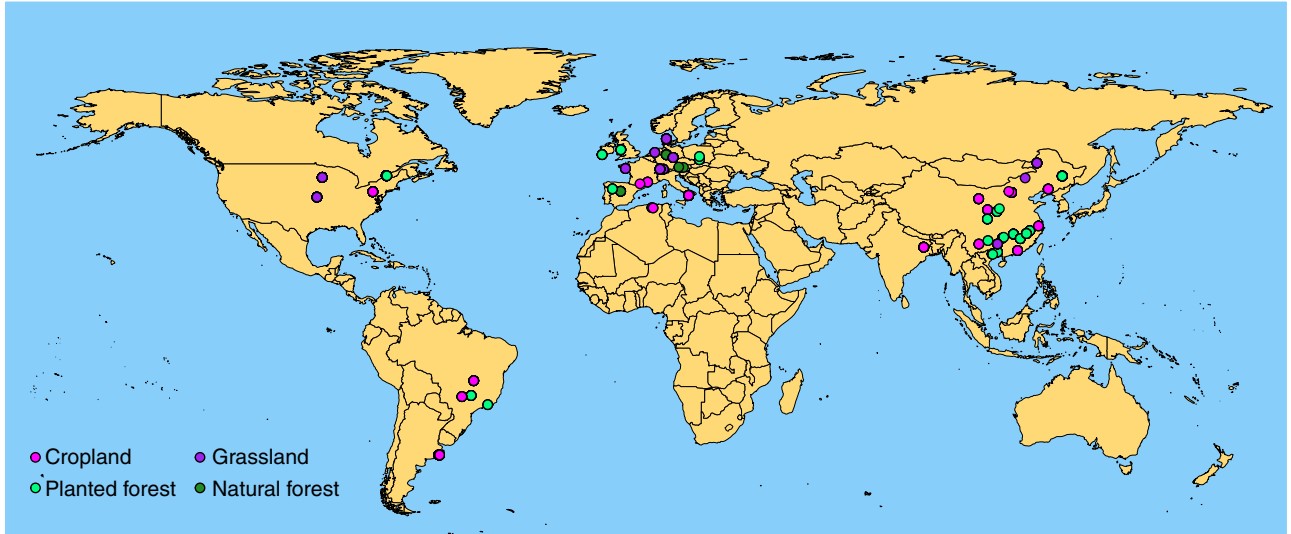

**Fig. 1** Global distribution of study sites in the meta-analysis. Magenta, purple, green and dark green points indicate study sites at croplands, grasslands, planted forests and natural forests. Experiments using planted containers were not included in this figure. Source data are provided as a Source Data file

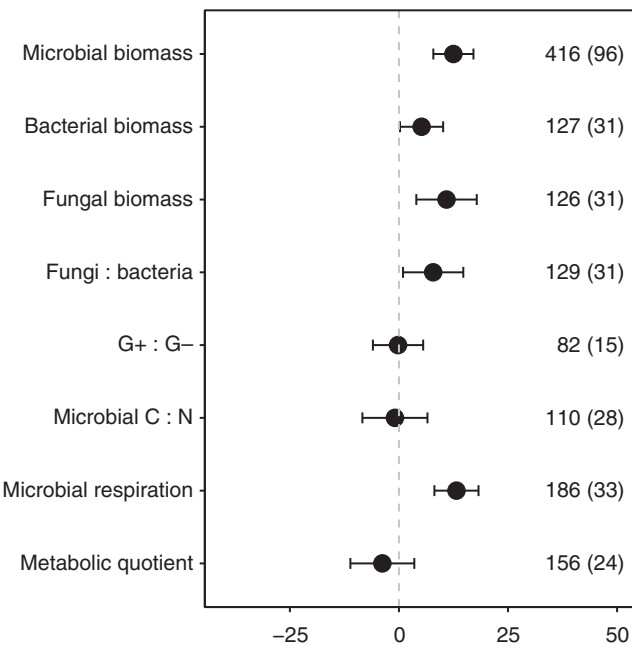

**Fig. 2** Comparison of soil microbial attributes in plant mixtures versus monocultures. The effects represent the increase or decrease (%) of a given microbial attribute compared to the corresponding mean of constituent monocultures at the mean species richness and mean stand age in mixtures (see Methods). Values are mean ± 95% confidence intervals of the percentage effects between the plant mixtures and monocultures. The number of observations is shown beside each attribute without parentheses with the number of studies in parentheses. G+:G− represents Gram-positive bacteria and Gram-negative bacteria biomass ratio. Source data are provided as a Source Data file

**The variation of plant mixture effects**. With increasing species richness in plant mixtures, the effect sizes for microbial biomass, bacterial biomass, fungal biomass and microbial respiration significantly increased ($P < 0.001$, $P = 0.004$, $P = 0.008$ and $P < 0.001$, respectively; Fig. 3a, Supplementary Table 2). The effect size for G+:G− bacteria ratio decreased with the species richness

($P = 0.040$, Supplementary Table 2). The effect sizes for microbial biomass and respiration increased with stand age (both $P < 0.001$), and the effect size for fungi:bacteria ratio decreased ($P = 0.046$) (Fig. 3b, Supplementary Table 2). The increases of microbial biomass and respiration with the species richness became more pronounced in older stands (both $P < 0.001$), but not for other microbial attributes (Fig. 4, Supplementary Table 2).

The plant mixture effect on microbial attributes did not differ significantly among ecosystem types including natural forests, planted forests, planted grasslands, croplands, and planted containers (Supplementary Figure 1). For studies conducted under natural climates (forests and grasslands), the responses of microbial attributes to plant mixtures did not change significantly with mean annual temperature nor aridity index of study sites except microbial respiration. Both the average effect of plant mixtures and the effect of species richness in mixtures on microbial respiration were more pronounced in colder climates ($P < 0.001$ for both the average and interaction effects) (Supplementary Figure 2).

**Predicted responses of microbial biomass and respiration**. Predicted from the fitted species richness- and stand age-dependent responses (Fig. 4), a 10% decrease in plant species richness (from 100 to 90%) over one year reduced microbial biomass and microbial respiration by 5%. A 40% decrease in plant species richness (from 100 to 60%) over one year led to a 20% reduction in microbial biomass and microbial respiration (Fig. 5). The declines in microbial biomass and respiration in response to the decrease in plant richness became amplified with longer stand age (Fig. 5). For example, a 10% decrease in plant species richness (from 100 to 90%) over five years led to a 16% lower microbial biomass (Fig. 5).

## Discussion
Our results demonstrated that plant diversity increased soil microbial biomass across a diverse range of terrestrial ecosystems. Our results extend those derived from twelve grassland studies[40] to a diverse range of ecosystem types. Importantly, we showed that the plant diversity effect on microbial biomass increased logarithmically with the species richness in mixtures, which coincides with the pattern of the diversity-productivity relationship in the plant community[5,7]. For forests, a 10% decrease in tree

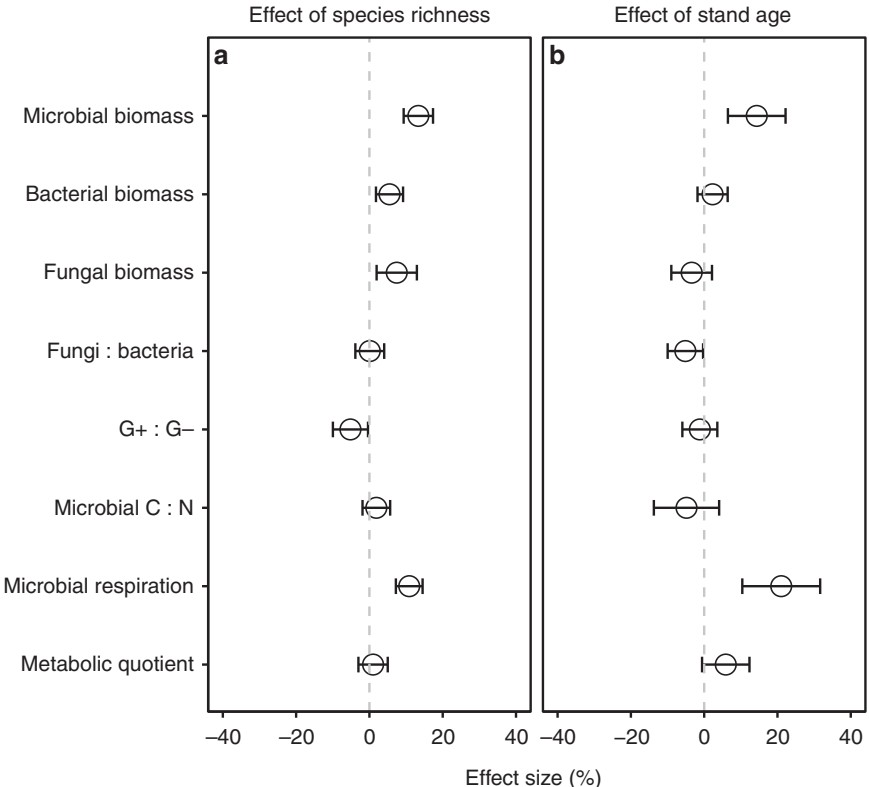

**Fig. 3** The effects of plant mixtures on microbial attributes in relation to plant species richness and stand age. **a** The plant species richness (log scale) in mixtures. **b** Stand age (years). The effects represent the estimated coefficients of the species richness in mixtures and stand age. Values (estimated $\beta_1$ and $\beta_2$ in Equation (3), respectively, see Methods) are mean ± 95% confidence intervals. G+:G− represents Gram-positive and Gram-negative bacteria biomass ratio. Source data are provided as a Source Data file

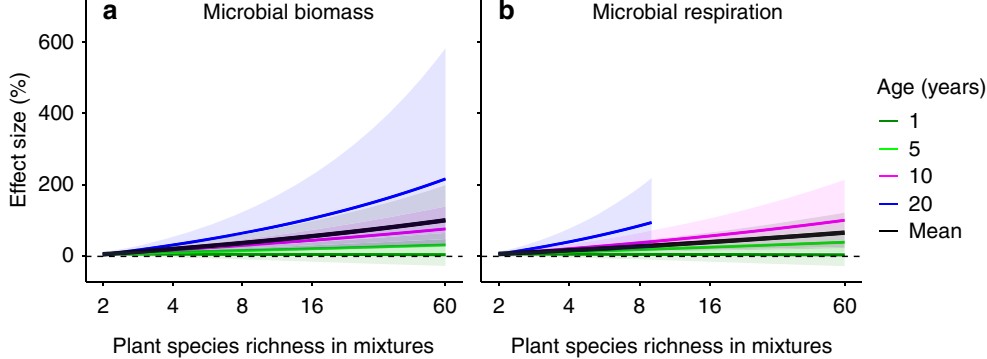

**Fig. 4** The interactive effects of the plant species richness in mixtures and stand age on microbial attributes. **a** Microbial biomass. **b** Microbial respiration. The effects are quantified as the percent changes in mixtures compared to the corresponding mean value of constituent monocultures. Lines are fitted age-dependent regressions with 95% confidence intervals in shade. Dark green, green, magenta, blue and black lines indicate stand age at 1, 5, 10 and 20 years and mean stand age across all observations. Source data are provided as a Source Data file

species richness (from 100 to 90%) is reported to cause a 2–3% decline in forest productivity[7]. For microbial communities, a 10% decrease in plant species richness (from 100 to 90%) over one year causes a 5% decline in microbial biomass, where the extent of the decline would be larger over the long term. Further, while previous experimental studies showed that plant diversity increased fungal but not bacterial biomass[21,22,141], our study offers global evidence that the abundance of both bacteria and fungi increase with plant species diversity. This may be attributable to that higher productivity induces more carbon and nutrient inputs to the soil in mixtures, benefitting both fungi and

bacteria, as well as the facilitative effect of fungi on the penetration of bacteria into leaf tissue[142].

Our study offers new insights into the variations of fungi:bacteria ratio and G+:G− bacteria ratio associated with plant species diversity. Although both bacteria and fungi biomass increased with plant diversity, fungi:bacteria ratio was higher in plant mixtures than in monocultures, suggesting that fungi may benefit more from plant species mixture, likely because fungi, but not bacteria, can transfer nutrients from high- to low-nutrient plant litter when plant mixtures include species with different nutrient contents[143,144]. Also, G+:G− bacteria ratio decreased in

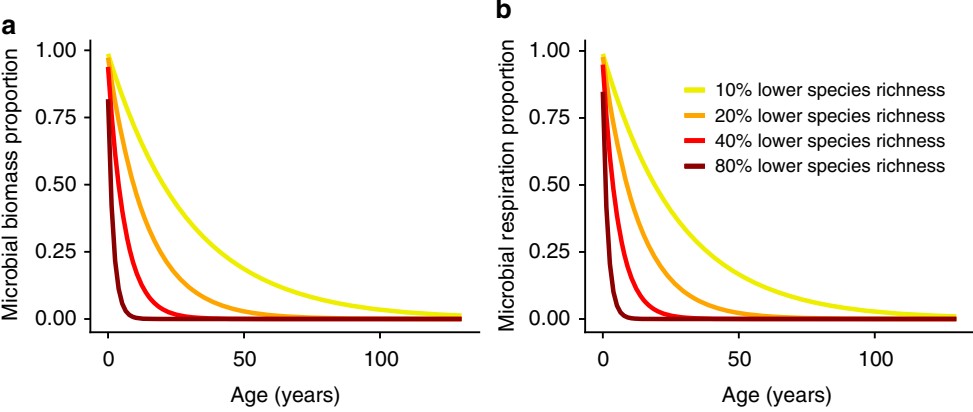

**Fig. 5** Predicted responses of microbial attributes to a range of plant species richness reductions. **a** Microbial biomass. **b** Microbial respiration. Yellow, orange, red and dark red lines indicate plant species richness reduction at 10, 20, 40 and 80%. Source data are provided as a Source Data file

mixtures, attributable to increased soil moisture with plant diversity[22], which promotes the relative abundance of G− bacteria[23,24]. In contrast to our hypothesis, we did not find an effect of plant mixtures on microbial C:N. This is likely because plant diversity increases soil carbon and simultaneously increases nitrogen retention[145]. Consistent with our hypothesis and previous findings[29,35], we found microbial respiration was higher in plant mixtures.

Our study also showed that the species mixture effects on microbial biomass and respiration increased with the species richness in mixtures and stand age with more pronounced species richness effects in older stands. The increases of soil microbial biomass and respiration with plant species richness were anticipated because soil carbon available for soil microorganisms increases with plant species richness[14,17,18]. The significant interaction effect of species richness and stand age is likely because soil microorganisms show a time lag in their response to plant community after stand establishment[34,35], and species-rich systems embrace a stronger increase of the plant-derived resources for soil microorganisms over time[32]. Together, our analysis indicates that the lack of plant diversity effects on soil microorganisms in certain studies is attributable to short experimental duration and limited plant species richness. Importantly, our finding indicates that the stronger time-dependent effects in species-rich systems do not only occur for ecosystem productivity[33] but also for soil microbial biomass and respiration. Overall, our study suggests a lasting and deepening effect of plant diversity on microorganisms with increasing stand age.

Although a wide range of ecosystem types were covered, including natural forests, planted forests, planted grasslands, croplands, and planted containers in this study, we found no significant effect of ecosystem types on soil microbial responses to plant mixtures except for a mean annual temperature-dependent response of soil microbial respiration to species mixture under natural climates. The increased species mixture effect on microbial respiration with decreasing mean annual temperature corroborates those of plant productivity to diversity along climate gradients[38,39]. Collectively, these results suggest that the decrease of ecosystem functions of plant and microorganisms with lower plant diversity is more pronounced in colder environments.

Our estimated effects of plant diversity on soil microbial attributes represent plant species complementarity since we factored out selection effects related to plant species composition, following the method by Loreau and Hector[146]. However, among the 106 original studies, 12 studies that reported time-dependent responses had only initial species compositions. Because species compositions can change over time, the log response ratios between plant mixtures and monocultures calculated based on initial species composition could include selection effects associated with temporal changes in species compositions[147]. Nevertheless, a long-term grassland experiment shows that the strength of the complementary effect increases while the selection effect tends to decrease over time[33]. Moreover, we note that previous analyses infer the effects of species diversity loss from experimental studies that manipulated species richness levels[6]. However, time-dependent responses to species diversity could differ between artificially and randomly assembled communities and species loss in natural ecosystems[148]. For this reason, we focus on interpreting the relation between plant species diversity and soil microbial attributes, rather than species loss.

We found that plant species mixtures increased soil microbial biomass, fungal to bacterial biomass ratio, and respiration across global terrestrial ecosystems. We further revealed that these effects increased with the number of species in mixtures and stand age. While the responses of soil microbial attributes to plant diversity corroborate with those of plant productivity, they were higher in magnitude than those of plant productivity. Moreover, our findings indicate that the increase of microbial respiration with plant diversity was more pronounced under a colder climate. Because of the dominant control of soil microorganisms on element cycling[1–4], our results suggest that declines in soil microbial biomass and respiration, coupled with the shift in the relative abundance of soil fungi and bacteria induced by lower plant diversity, could have profound adverse effects on the global carbon and nutrients cycles. These adverse effects could amplify with further decreases in plant diversity and longer stand age.

## Methods
**Data collection**. We systematically searched all peer-reviewed journal articles and theses that investigated the effects of plant diversity on microbial biomass, fungi: bacteria ratio, G+:G− bacteria ratio, microbial C:N ratio and respiration, using the Web of Science and Google Scholar, up to 1st November, 2018. The literature search was performed following guidelines from PRISMA (Preferred Reporting Items for Systematic Reviews and Meta-Analyses;[149] Supplementary Figure 3). Various keyword combinations were used for the search, such as (species diversity OR tree diversity OR richness OR mix species OR mix tree OR mixture OR intercrop) AND (microbial abundance OR microbial biomass OR microbe OR microbial OR soil biota OR microbial community OR fungi OR bacteria OR microbial biomass nitrogen OR microbial respiration OR basal respiration OR microbial activity). We reviewed each article to determine whether the studies met the following criteria: (1) isolated the effects of plant diversity from other factors, such as stand age and topography; (2) the microbial attributes could be extracted directly from the text, tables, and figures.

When different publications included the same data from one study, we recorded the data only once. When a study included plant mixtures of different numbers of species, we considered them distinct observations. Also, when a publication included several experiments under different abiotic conditions, such as different locations, treatments, stand ages, and soil layers, we considered them different observations. We used Plot Digitizer version 2.0 (Department of Physics at the University of South Alabama, Mobile, AL, USA) to digitally extract data from figures when the results were graphically reported. We obtained a meta-dataset of 1332 observations from 106 studies (Supplementary Table 1) that involved the mixing of live plants in natural forests, planted forests, planted grasslands, croplands, and planted containers (Fig. 1).

For each study, we extracted microbial attributes including microbial biomass, bacterial biomass, fungal biomass, fungi:bacteria ratio, G+:G− bacteria ratio, microbial C:N ratio, microbial respiration, and metabolic quotient. Methods for determining the microbial biomass included the measurement of microbial biomass carbon via a substrate-induced respiration method[150], fumigation-extraction method[151], measuring the total amounts of phospholipid fatty acid (PLFA) in the soil[152], as well as the investigation of microbial quantities via a quantitative polymerase chain reaction (qPCR). Metabolic quotient was calculated as the respiration rate per unit microbial biomass.

We obtained plant species richness, species identity, the species proportions in plant mixtures, geographical location (latitude, longitude, and altitude), ecosystem type, stand age, and soil sampling depth (as middle value of each sampling depth interval[153]) from original publications. The species proportions in plant mixtures were based on basal area, stem density, or crown cover in forests, coverage or sowing seeds in grasslands, and the number of individuals in other systems. In 3 of the 106 studies, species proportions were unavailable in the publications, and we assumed they were equal for all constituent species. The stand age referred to the years between stand establishment or experiment initiation and the measurement of soil microbial attributes. For those studies conducted under natural climates including forests and grasslands, we obtained mean annual temperature for each study site from original publications; when unavailable, it was based on geographic locations using the WorldClim version 2 dataset[154]. Similarly, we derived the aridity index (calculated as mean annual precipitation divided by mean annual potential evapotranspiration, where a higher aridity index indicated lower aridity) for each site from the Global Aridity and PET Database[155].

**Data analysis**. We used a natural log-transformed response ratio (lnRR) as the effect size to assess the responses of soil microbial attributes to plant mixtures[156]. The lnRR was calculated as:

$$\ln RR = \ln(X_t/X_c) \tag{1}$$

where $X_t$ and $X_c$ are the observed and expected values in a mixture, respectively. To take account of the species compositional effect (or selection effect), we calculated the $X_c$ as the weighted values of the constituent species in monocultures following Loreau and Hector[146], in which weights represent the species proportions in the mixture. Therefore, our estimated lnRR represents the complementarity effect of species mixtures. Within each study, lnRR was calculated separately for each species diversity level and stand age.

In the meta-analyses, effect size estimates and subsequent inferences may be dependent on how the individual observations are weighted. Weightings based on sampling variances might assign extreme importance to a few individual observations, and consequently, the average lnRR would be mainly determined by a small number of studies. Similar to the previous studies[31,157], we used the number of replications for weighting:

$$W_r = (N_c \times N_t)/(N_c + N_t) \tag{2}$$

where $W_r$ is the weight for each observation and $N_t$ and $N_c$ are the numbers of replications in the plant mixtures and the corresponding monocultures, respectively.

For each microbial attribute, we tested whether its response to plant mixtures differed from zero and whether lnRR was affected by plant species richness ($R$), stand age ($A$, years), and ecosystem type ($E$) using the following model:

$$\ln RR = \beta_0 + \beta_1 \cdot R + \beta_2 \cdot A + \beta_3 \cdot R \times A + \beta_4 \cdot E + \beta_5 \cdot R \times E + \beta_6 \cdot A \times E \\ + \beta_7 \cdot R \times A \times E + \pi_{\text{study}} + \epsilon \tag{3}$$

where $\beta$ is the coefficient to be estimated; $\pi_{\text{study}}$ is the random effect factor of study, accounting for the autocorrelation among observations within each study; $\varepsilon$ is sampling error. We conducted the analysis using restricted maximum likelihood estimation with the *lme4* package with $W_r$ as the weight for each corresponding observation[158]. To prevent overfitting[159], we selected the most parsimonious model among all alternatives with the condition to keep $R$ and $A$, as they were part of our core hypotheses to be tested. The model selection was accomplished by using the '*dredge*' function of the *MuMIn* package[160]. All terms associated with ecosystem type (Equation 3) were excluded in the most parsimonious models (Supplementary Table 2). To further examine the effects of ecosystem types, we conducted an analysis with the ecosystem type as the only fixed factor and study as the random factor, and the analysis confirmed that there was no difference in the microbial responses among ecosystem types (Supplementary Figure 1). We assessed the assumption of linearity between lnRR and continuous predictors by comparing

linear and log-linear responses. Natural log transformed $R$, ln($R$), yielded lower or similar Akaike information criterion (AIC) values than $R$, whereas $A$ was better than ln($A$) (Supplementary Table 3).

We scaled all continuous predictors (observed values minus mean and divided by one standard deviation). When continuous predictors are scaled, $\beta_0$ is the overall mean lnRR at the mean ln($R$) and mean $A$[161]. To graphically illustrate whether the effect of the species richness in mixtures on lnRR differed with stand age, we calculated age-dependent effects using the recommended method[161] at stand ages of 1, 5, 10, and 20 years, respectively. While the stand age of our dataset ranged to 130 years, the trend for the stand age over 20 years was not shown since only two species mixtures and corresponding monocultures were studied for stands older than 20 years.

For studies conducted under natural climates including forests and grasslands, we examined whether mean annual temperature and aridity index of study sites affected the responses of microbial attributes to the effects of the species richness in the mixtures and stand age by substituting $E$ in Equation (3) by mean annual temperature and aridity index, respectively. Similarly, we selected the most parsimonious models using the method described above. All terms associated with mean annual temperature and aridity index were excluded during the model selection except for microbial respiration. The response of microbial respiration to plant mixtures was mean annual temperature-dependent (Supplementary Figure 2).

For ease of interpretation, lnRR and its corresponding 95% confidence intervals (CIs) were transformed to a percentage change between monocultures and mixtures as

$$(e^{\ln RR} - 1) \times 100\% \tag{4}$$

i.e., overyielding from species complementarity[146]. If the CIs did not cover zero, the effect of species mixture on microbial attributes differed significantly at $\alpha = 0.05$ between monocultures and mixtures.

To illustrate the effects of plant diversity on microbial biomass and respiration over time, we compared the lnRR when the plant richness in mixtures was $R_1$ (all species present) and $R_\alpha$ (α% lower species richness). We assumed that the mean value of monocultures, $X_c$, did not vary with the number of monocultures of different species, which led to the following equation:

$$P_\alpha = (R_\alpha/R_1)^{\beta_1 + \beta_3 \cdot T} \tag{5}$$

where $P_\alpha$ is the proportion of remaining microbial biomass or respiration under α% lower plant species richness in a period of $T$, and other model terms were described in Equation (3). The detailed derivation process for Equation (5) is presented in Supplementary Methods. Based on Equation (5), we fitted curves for the decrease in microbial biomass and respiration over time when there was a 10, 20, 40, and 80% decrease in plant species richness.

We examined whether the responses to plant mixtures differed with the technical methods for the determination of microbial biomass, bacterial biomass, fungal biomass, and fungal:bacterial biomass ratio. The technical methods showed insignificant effects on the effect size of plant mixture effects on microbial attributes (Supplementary Table 4). We also tested whether the responses to plant mixtures differed with soil depth, and we found consistent responses across all sampling depths (Supplementary Table 5). All analyses were performed in R 3.5.2[162].

**Reporting summary**. Further information on experimental design is available in the Nature Research Reporting Summary linked to this article.

## Code availability
The R scripts needed to reproduce the analysis is available as Supplementary Software.

## Data availability
The source data underlying Figs. 1–5 and Supplementary Figures 1–2 and Supplementary Tables 1–5 are provided as a Source Data file.

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

## Acknowledgements

We thank Mr. Eric Searle for assisting with data analysis, and Dr. Markus Lange for providing his experimental data in The Jena Experiment and the funding agency, German Research Foundation (DFG). This study was funded by the Natural Sciences and Engineering Research Council of Canada (RGPIN-2014–04181, RTI-2017–00358, STPGP428641, and STPGP506284) and National Science Foundation of China for Distinguished Young Scholars (31625007).

## Author contributions

C.C. and H.Y.H.C. designed research; C.C. and X.C. collected data; C.C., H.Y.H.C., and X.C. analysed data; C.C., H.Y.H.C., X.C., and Z.H. wrote the article.

## Additional information

**Competing interests:** The authors declare no competing interests.

