## [Peer Review File · Nature Communications]

Reviewers' Comments:

Reviewer #1:

Remarks to the Author:

This paper investigates the relation between plant species diversity and soil microbial parameters (microbial biomass, soil respiration, bacterial biomass, fungal biomass, fungi:bacteria ratio, ratio of gram-positive to gram-negative bacteria, microbial C:N ratio and metabolic quotient). The authors perform a meta-analysis, using 1150 paired observations (monocultures-mixtures) from 103 published studies. They conclude that plant diversity loss reduced some of the parameters, across terrestrial ecosystems, and that the magnitude of the effect would be linked to the extent of species loss, time, and environmental stress. Given the relatively fewer information on soil microbial parameters in relation to aboveground species diversity, compared to aboveground parameters, the study is of great interest. But I found the description of the statistics and the discussion of the results highly incomplete, so that I am not convinced of the validity of the findings reported.

Below I detail some general concerns (G), followed by specific concerns relating to line numbers in the manuscript.

G1: In the title, and at several places within the text, authors claim to find effects of 'biodiversity loss'. As the study uses data from experimental as well as observational data, 'diversity loss' was not studied. One can only infer the relation between diversity and parameters measured.

G2: The description of the selection of publications is limited to the keywords used and the fact that the papers studied plant diversity and that data could be extracted. Also it is indicated that studies 'involved the mixing of live plants in natural forests, planted forests, planted grasslands, croplands, and containers'. I guess that this is called 'experimental system type' in the text and 'ecosystem' in S2. It should be made clear (and be coherent) across the whole manuscript what is what. Further information should be provided on:

Number of data for each ecosystem type

Plant diversity levels present in these studies, and the ranges for each ecosystem type

How the authors dealt with different diversity levels within one study (would one study with monocultures and 4 additional diversity levels lead to 4 data pairs per replicate? – this would lead to the use of the same monoculture data several times – is this a problem?)

Discuss the implication of use of different approaches (experimental, observational, different mixing patterns) for interpretation of results

I wonder how the relative distribution of mixture levels might affect the results (for age: over 20 years only monocultures and two species mixtures)

G3 Statistics

I am surprised that ecosystem type is not included in the model testing the tree species richness effects, but tested separately (equations (3) and (4)). I guess that data from lower species richness are mainly derived from forests and the high species richness data (up to 60!) are from grassland studies. It seems strange to derive relationships across ecosystem types, without including this parameter in the model. Also, information on model fit should be provided. Even if some relations are significant, the scope of the results is different if the model explains 90% of the variability of the data rather than 1%.

G4 Discussion: overall I find that the discussion does not provide an in depth insight of the significance of the results. Mainly findings of the study are cited and related to some literature data supporting the findings. Important issues of biodiversity-functioning topic and limitations of this

approach are ignored. For example: tree species composition, complementarity/selection effect (depending on the experimental design from which the data is derived), definition of 'diversity', the issue on functional diversity.

Abstract

L 11: As the term 'microbe' is generally associated with disease (definition Oxford dictionary: 'A microorganism, especially a bacterium causing disease or fermentation'), I would suggest to use the term 'microorganism' throughout the manuscript

L11 : 'material cycling' → 'biogeochemical cycling' (suggestion)

L13 : 'impact' : replace by a more appropriate term throughout the manuscript (see <https://schimelwritingscience.wordpress.com/2018/04/29/protect-verbdiversity-or-why-i-hate-impact-redux/>)

L 15 : 'composition' be more precise

L 15 'respiration' → 'soil respiration'

L16-20: I find that the terms 'negative impact' and 'positive impact' are not specific enough. A 'negative impact' would imply deleterious (bad) consequences, which could be related to decreases or increases in parameters. For example an increase in soil exchangeable aluminum is not desirable, but how would it be described, negative or positive impact? I therefore suggest to describe the effects in terms of increase or decrease.

L 20: difficult to read, and see comments above

L 22-25: this conclusion makes too great generalizations and should be revised

Introduction

L29: 'which' wrong use

L32: 'Plant diversity loss affects soil microbes negatively': this is a big generalization, and the effect of plant diversity loss depends on the species present (if you remove a species secreting inhibitors, microbial parameters might be improved). Also it should be mentioned what parameters are expected to be affected 'soil microbes' is too general (does one think on biomass, composition activity?)

L38 ', others have found that plant diversity has no, or even negative effects, on soil microbial biomass' this is contradictory to L32-33, and again depends on environmental context and species present

L43-44 The study cited involves functionally different grassland species. When functionally similar species are involved, it is not sure that mixing would lead to higher root exudate diversity

L97-98: 'the effect of plant diversity loss would increase with environmental stress and soil sampling depth'. This is not really investigated in the study, there is no data on this in the result section, and only one parameter each is significantly related to MAT or aridity index (supplementary material, but main factors seem not included in the model). Further I believe extending MAT or aridity to 'environmental stress' goes too far.

Discussion

L 180 'we did not find positive effects of plant diversity on microbial C:N'. What would be a positive effect on microbial C:N? This ratio reflects the characteristics of the microorganism present, so a change is likely to indicate a change in the relative importance of microorganisms present (bacteria/fungi).

Data analysis

L 264: The authors report the calculation of a 'species ratio of plant mixtures', it is not clear to me what this is (a ratio is a relationship between two quantities, a quotient of two mathematical expressions). Furthermore, the next sentence indicates that the calculation of species ratio was not possible, and so the species richness was used for the whole study. This paragraph should be clarified.

Also in the tables and figures, and throughout the text, the same term should be used (Fig. 1: plant diversity effects, others: species richness in axis and plant mixture in the legend).

L268: Start a new paragraph for the response ratio, to separate between the independent variable (first paragraph) and the dependent variables

L272and following:the information of weightings is not clear. Were only monoculture data weighted? Were values multiplied by W_r ?

L282: what method and program were used? Please indicate how and why the five alternative models were selected

L 292 'When continuous predictors are scaled...'. I guess that they were scaled in this study, so this information should be provided earlier and reworded ('continuous predictors $\ln(R)$ and T were scaled...')

L301 Although I guess that this approach might be mathematically correct, I have major concerns about the biological validity. The dataset contains data on soil microbial biomass and respiration from monocultures and mixtures of highly different plants (grasslands, forests) measured at different stand ages. I do not believe that a global decrease in plant species richness is a meaningful measure for such different species. Further, the stand age (of the data from different studies) does not reflect the time during which a change in microbial biomass would occur. Even within the same ecosystem type i.e. forest, if you have microbial biomass from a monoculture of 5 years age and from a mixture of 10 years, one cannot conclude that the difference in microbial biomass between the two would be the same than if you remove species from a mixture and wait 5 years. This is even worse if you mix data from grasslands and forests and where species richness values might be linked to ecosystem type (see G3).

L312: environmental factors are added to the equations, but it looks as if the main effects $\ln(R)$ and T were not included, as they do not appear in Table S2. This approach (when including the main effects) seems to make the use of equations (3) and (4) not necessary. I could not follow the reasoning of the rest of the paragraph.

Supplementary

Provide all equations in the supplementary file, some equations are in the paper, others are in supplementary material, which makes reading and understanding awkward

Reviewer #2:

Remarks to the Author:

Here Chen et al. presents an interesting dataset that summarizes the effects of plant diversity on microbial biomass, respiration and potential function. In light of the recent IPCC report a study that relates plant diversity to carbon in the soil at a global scale may be of particular importance. Using meta-analyses techniques the authors are able to bring together data from over 1000 observations from 100+ studies, an impressive feat. However, the presentation of the data is insufficient whereby only relations between plant diversity and biomass are shown, while the impacts of these relationships are not assessed or discussed. Without addressing the impacts of the compelling relationships, the paper seems unfinished. At the very least a discussion and assessment by ecosystem needs to be added.

Specific Comments:

Overall writing: while the text is good, many sentences do not relate well – one to the other. This is not an issue of grammar, but rather the text does not help to illustrate the study and why it is novel.

Title: 'Global negative impacts'? ...

Introduction: needs to be condensed and focused. Currently it is written more as a review of all studies but does not synthesize that information in a way to prepare the reader for the results. Needs to better highlight what has been done, what needs to be done, and why this is important. Further, the hypotheses are more predictions and don't get at the mechanisms for why understanding relationships between plant diversity and soil biomass/respiration are important.

Results:

- Needs analysis by ecosystem or other global factors. Calculations of how much carbon this accounts for would also be really interesting and could add a new dimension to the study. Or at least better explain if/how the loss of biodiversity is meaningful to an ecosystem.
- A map of locations would be helpful
- Throughout it needs to be made clear that this is potential function.
- By experiment type – need to show that they responded the same

Reviewer #3:

Remarks to the Author:

I read the manuscript by Chen et al. on how loss of plant diversity detrimentally affects soil microbial biomass and some other soil microbial parameters. In general, I liked the manuscript and the efforts of the authors to compile such a database is praiseworthy. I, however, also have a number of issues with the presentation of the results and thus the overall message of the paper. I have detailed those below.

The main results (Fig. 1 and 2) presented by the authors are difficult to follow. First, how results in Fig. 1 are different compared to Fig. 2 (left panel) except that the units of effect size are different. Second, if authors have information on species richness differences among studies, why not use those differences as a covariate/moderate to explain the variation in effect size which is shown in Fig. 1. Third, I am unsure whether the stand age applies to all studies given some studies are from grasslands and/or greenhouse experiment. Thus, using this as a proxy of time or duration of the experiment is confusing and needs to be clarified. And, finally clarification on stand age will help understand Fig. 3 and the results shown in Table S1, such as the significant interaction between stand age and species richness on two microbial parameters.

I appreciate the predictive approach used by the authors (Fig. 4), but this is problematic in the sense that environmental factors could not be included in this. Authors have themselves shown that certain environmental factors explain the variations in plant diversity effects on microbial parameters (Fig. S1, Table S4). Could we inform these statistical models with environmental parameters? At least, this needs to be discussed in the result and the discussion.

The writing needs to be improved for clear transition. For instance, the introduction section lacks providing underlying reasons for why we should expect plant diversity effects on soil microbial parameters. I have provided some specific examples in my other comments below.

Other comments

Line 14: What does paired observation mean? Does it mean one monoculture vs. one mixture? I am not sure. Please clarify.

Line 18: gram-positive to gram-negative bacterial ratio.

Line 20: "the extent of biodiversity loss" is quite vague. Please clarify.

Line 22: in ecosystems from colder climates.

Line 34: by providing

Line 49: The abbreviation G+: G- ratio should be G+: G- bacteria ratio.

Line 59: Metabolic quotient comes without a proper transition. Either provide some background before the term is introduced or begin with a new paragraph here.

Line 63: microbial C:N ratio.

Lines 68-73: I could not get the underlying reason why there is time lag in plant diversity effect on microbial biomass. Please clarify.

Lines 80-82: Unclear. Why under environmental stress, plant diversity would have higher positive interspecific plant interactions than plant monocultures?

Line 90: Here, we conducted...

Line 97: I don't think authors have developed an adequate background in the introduction for "the extent of diversity loss" effects on soil microbial communities. This needs to be properly developed for readers to understand what authors mean by the extent of biodiversity loss, why this could be important for soil microbial communities and how their study aims to tackle this.

Line 101: Which microbial biomass is this..C or N?

Line 108-109: Why is this result relevant? And why not also run correlation for other response variables?

Editor's comments

R: We have prepared our resubmission based on your instructions. We thank the three anonymous reviewers for the constructive and insightful comments, which have been greatly helpful to improve our manuscript substantially.

Reviewer #1 (Remarks to the Author):

This paper investigates the relation between plant species diversity and soil microbial parameters (microbial biomass, soil respiration, bacterial biomass, fungal biomass, fungi:bacteria ratio, ratio of gram-positive to gram-negative bacteria, microbial C:N ratio and metabolic quotient). The authors perform a meta-analysis, using 1150 paired observations (monocultures-mixtures) from 103 published studies. They conclude that plant diversity loss reduced some of the parameters, across terrestrial ecosystems, and that the magnitude of the effect would be linked to the extend of species loss, time, and environmental stress. Given the relatively fewer information on soil microbial parameters in relation to aboveground species diversity, compared to aboveground parameters, the study is of great interest. But I found the description of the statistics and the discussion of the results highly incomplete, so that I am not convinced of the validity of the findings reported.

R: Please find our detailed responses to your concern below

Below I detail some general concerns (G), followed by specific concerns relating to line numbers in the manuscript.

G1: In the title, and at several places within the text, authors claim to find effects of 'biodiversity loss'. As the study uses data from experimental as well as observational data, 'diversity loss' was not studied. One can only infer the relation between diversity and parameters measured.

R: Our original thought was that our study was set in a similar context as *Cardinale et al. (2012)*, in which most data are from experiments that manipulated species diversity. In the recent paper (*Kardol et al. 2018*), plant diversity loss was actually manipulated. With this new understanding, we have reworked our text throughout to improve clarity.

G2: The description of the selection of publications is limited to the keywords used and the fact that the papers studied plant diversity and that data could be extracted. Also it is indicated that studies 'involved the mixing of live plants in natural forests, planted forests, planted grasslands, croplands, and containers'. I guess that this is called 'experimental system type' in the text and 'ecosystem' in S2. It should be made clear (and be coherent) across the whole manuscript what is what. Further information should be provided on:

R: Great point about changing ecosystem type to experimental system type. We have made this change throughout.

Number of data for each ecosystem type

R: We have added **Fig. S1 to show the number of data for each experimental system and corresponding text results.**

Plant diversity levels present in these studies, and the ranges for each ecosystem type

R: We have added **Fig. S1 to show the plant species richness in each experimental system type for each microbial attribute.**

How the authors dealt with different diversity levels within one study (would one study with monocultures and 4 additional diversity levels lead to 4 data pairs per replicate? – this would lead to the use of the same monoculture data several times – is this a problem?)

R: We are sorry for the lack of clarity. We have revised the corresponding text to improve clarity (L272–279). If a study has multiple levels of species richness (for example, 1, 4, 8, and 16), InRR was calculated for the species richness level 4, 8, and 16, respectively. We controlled autocorrelation among these levels within each study by using “study” as the random effect in our model. The same procedure was applied for studies with repeated measures over time (stand age effect).

Discuss the implication of use of different approaches (experimental, observational, different mixing patterns) for interpretation of results

R: Since we have found that experimental system has no effect on the influence of plant diversity on soil microbial abundance, composition, and function, we further emphasize this in Discussion (L167-168).

I wonder how the relative distribution of mixture levels might affect the results (for age: over 20 years only monocultures and two species mixtures)

R: Data limitation clearly exists when we examine the global variation associated with the species richness in mixtures and stand age. We have not attempted to speculate a particular data limitation.

G3 Statistics

I am surprised that ecosystem type is not included in the model testing the tree species richness effects, but tested separately (equations (3) and (4)). I guess that data from lower species richness are mainly derived from forests and the high species richness data (up to 60!) are from grassland studies. It seems strange to derive relationships across ecosystem types, without including this parameter in the model. Also, information on model fit should be provided. Even if some relations are significant, the scope of the results is different if the model explains 90% of the variability of the data rather than 1%.

R: In our original submission, we did include experimental system (but incorrectly called Ecosystem) in our model, but it appeared too late. We feel that we communicated this poorly in our original submission. In this revision, we have carefully revised the statistical method section to ensure our thought process is logic (L288–305).

G4 Discussion: overall I find that the discussion does not provide an in depth insight of the significance of the results. Mainly findings of the study are cited and related to some literature data supporting the findings. Important issues of biodiversity-functioning topic and limitations of this approach are ignored. For example: tree species composition, complementarity/selection effect (depending on the experimental design from which the data is derived), definition of ‘diversity’, the issue on functional diversity.

R: Sorry for our ambiguous writing in our original submission. The way we calculated InRR already accounted for selection effect (see revised statistical analysis, L275–279). Therefore, positive InRR resulted from complementarity. We recognized the importance of functional diversity, but we only have data for species diversity.

Abstract

L 11: As the term ‘microbe’ is generally associated with disease (definition Oxford dictionary: ‘A microorganism, especially a bacterium causing disease or fermentation’), I would suggest to use the term ‘microorganism’ throughout the manuscript

R: Thank you for this insight. We have replaced the respective words as recommended.

L11 : ‘material cycling’ → ‘biogeochemical cycling’ (suggestion)

R: Corrected as recommended (L12).

L13 : 'impact' : replace by a more appropriate term throughout the manuscript
(see <https://schimelwritingscience.wordpress.com/2018/04/29/protect-verbidiversity-or-why-i-hate-impact-redux/>)

R: Replaced throughout as recommended

L 15 : 'composition' be more precise

R: Later, we mentioned fungi:bacteria ratio and gram-positive to gram-negative ratio. We think the brief form here would flow better.

And L 15 'respiration ' → 'soil respiration'

R: here we indeed meant soil microbial respiration since soil respiration would also include autotrophic component. No change was made.

L16-20: I find that the terms 'negative impact' and 'positive impact' are not specific enough. A 'negative impact' would imply deleterious (bad) consequences, which could be related to decreases or increases in parameters. For example an increase in soil exchangeable aluminum is not desirable, but how would it be described, negative or positive impact? I therefore suggest to describe the effects in terms of increase or decrease.

R: Great, revised as recommended (L16–21).

L 20: difficult to read, and see comments above

R: Revised to improve clarity (L20–21).

L 22-25: this conclusion makes too great generalizations and should be revised

R: Revised by changing “indicated” to “suggests” and reworded others as well (L21–24).

Introduction

L29: 'which' wrong use

R: Deleted (L30).

L32: 'Plant diversity loss affects soil microbes negatively': this is a big generalization, and the effect of plant diversity loss depends on the species present (if you remove a species secreting inhibitors, microbial parameters might be improved). Also it should be mentioned what parameters are expected to be affected 'soil microbes' is too general (does one think on biomass, composition activity?)

R: Revised to be more specific.

L38 ' , others have found that plant diversity has no, or even negative effects, on soil microbial biomass' this is contradictory to L32-33, and again depends on environmental context and species present

R: We changed “is expected to” “may” so that the sentence is presented as a hypothesis (L30–31).

L43-44 The study cited involves functionally different grassland species. When functionally similar species are involved, it is not sure that mixing would lead to higher root exudate diversity

R: We have reworded the sentence (L41)

L97-98: 'the effect of plant diversity loss would increase with environmental stress and soil sampling depth'. This is not really investigated in the study, there is no data on this in the result section, and only one parameter each is significantly related to MAT or aridity index (supplementary material, but main factors seem not included in the model). Further I believe extending MAT or aridity to 'environmental stress' goes too far.

R: We have thought hard about this during the revision. At the end, we have deleted the referred text.

Discussion

L 180 'we did not find positive effects of plant diversity on microbial C:N'. What would be a positive effect on microbial C:N? This ratio reflects the characteristics of the microorganism present, so a change is likely to indicate a change in the relative importance of microorganisms present (bacteria/fungi).

R: Revised to improve clarity (L188–189).

Data analysis

L 264: The authors report the calculation of a 'species ratio of plant mixtures', it is not clear to me what this is (a ratio is a relationship between two quantities, a quotient of two mathematical expressions). Furthermore, the next sentence indicates that the calculation of species ratio was not possible, and so the species richness was used for the whole study. This paragraph should be clarified. Also in the tables and figures, and throughout the text, the same term should be used (Fig. 1: plant diversity effects, others: species richness in axis and plant mixture in the legend).

R: Sorry for the poor wording. We have revised the text accordingly (L260–262, L275–279) and figure captions (Figs. 2, 3, 4 and 5)

L268: Start a new paragraph for the response ratio, to separate between the independent variable (first paragraph) and the dependent variables

R: We have now moved the independent variables to L257–270.

L272 and following: the information of weightings is not clear. Were only monoculture data weighted? Were values multiplied by W_r ?

R: The values in monocultures and mixtures are used to calculate $\ln RR$, and $\ln RR$ was then as the dependent variable. Weighting in lmer models was specified as $weights = W_r$ as part of the model. We have added this in the text (L294–295)

L282: what method and program were used? Please indicate how and why the five alternative models were selected

R: In our original submission, the five alternative models were chosen based on our hypotheses developed in Introduction. During this revision, we have changed to "selected the most parsimonious model among the all alternatives with the condition to keep R and A as they were part of our core hypotheses to be tested" (see revised L295–302)

L 292 'When continuous predictors are scaled...'. I guess that they were scaled in this study, so this information should be provided earlier and reworded ('continuous predictors $\ln(R)$ and T were scaled...')

R: Revised as recommended (L306–308).

L301 Although I guess that this approach might be mathematically correct, I have major concerns about the biological validity. The dataset contains data on soil microbial biomass and respiration from monocultures and mixtures of highly different plants (grasslands, forests) measured at different stand ages. I do not believe that a global decrease in plant species richness is a meaningful measure for such different species. Further, the stand age (of the data from different studies) does not reflect the time during which a change in microbial biomass would occur. Even within the same ecosystem type i.e. forest, if you have microbial biomass from a monoculture of 5 years age and from a mixture of 10 years, one cannot conclude that the difference in microbial biomass between the two would be the same than if you remove species from a mixture and wait 5 years. This is even worse if you mix data from grasslands and forests and where species richness values might be linked to ecosystem type (see G3).

R: Sorry for the lack of clarity in our original submission. Species richness was meant to be the species richness in mixtures. Because we calculated InRR as the log response ratio of the value in a given plant mixture to the mean value of the corresponding monocultures, we have standardized the mixture effect within each level of the species richness in the mixture and age within each study (see clarified L275–279). Our core hypotheses are whether the InRR would increase with the species richness in mixture and over time, and whether these responses are dependent on experimental systems or in natural systems, dependent on MAT and AI.

L312: environmental factors are added to the equations, but it looks as if the main effects In(R) and T were not included, as they do not appear in Table S2. This approach (when including the main effects) seems to make the use of equations (3) and (4) not necessary. I could not follow the reasoning of the rest of the paragraph.

R: Yes, In(R) and T were included along with environmental factors. Apparently, we did not communicate well. We have now thoroughly rewritten data analysis section to improve clarity (L288–305)

Supplementary

Provide all equations in the supplementary file, some equations are in the paper, others are in supplementary material, which makes reading and understanding awkward

R: We have now streamlined our presentation and only one equation (Eqn. 3) is used for hypothesis testing in main text.

Reviewer #2 (Remarks to the Author):

Here Chen et al. presents an interesting dataset that summarizes the effects of plant diversity on microbial biomass, respiration and potential function. In light of the recent IPCC report a study that relates plant diversity to carbon in the soil at a global scale may be of particular importance. Using meta-analyses techniques the authors are able to bring together data from over 1000 observations from 100+ studies, an impressive feat. However, the presentation of the data is insufficient whereby only relations between plant diversity and biomass are shown, while the impacts of these relationships are not assessed or discussed. Without addressing the impacts of the compelling relationships, the paper seems unfinished. At the very least a discussion and assessment by ecosystem needs to be added.

R: In our original submission, we did include analysis for ecosystem type effect, which is now called the effect of experimental system type, as recommended by Reviewer 1. We have now streamlined our analysis (L288–302) and presented the result (Fig. S1, Table S1)

Specific Comments:

Overall writing: while the text is good, many sentences do not relate well – one to the other. This is not an issue of grammar, but rather the text does not help to illustrate the study and why it is novel.

R: We have carefully revised all text by keeping the novelty issue in mind (L86–89).

Title: 'Global negative impacts'? ...

R: Revised

Introduction: needs to be condensed and focused. Currently it is written more as a review of all studies but does not synthesize that information in a way to prepare the reader for the results. Needs to better highlight what has been done, what needs to be done, and why this is important. Further, the hypotheses

are more predictions and don't get at the mechanisms for why understanding relationships between plant diversity and soil biomass/respiration are important.

R: We intended to use softer language to say the importance of our analysis. We feel that we have adequately highlighted our novelty (L86–89). We acknowledged previous work (L83–86). We developed our hypothesis 1 (i.e., prediction, L48–49) by reviewing the current mechanistic understanding (L38–47), hypothesis 2 (L59–61) similarly by synthesizing literature (L50–59), and hypothesis 3 (L70–71). Lastly, we postulated the idea that the plant diversity effect on soil microorganisms may change with experimental systems and environment (L72–82). We hope that our writing is effective for what we intended to accomplish. We recognize that we can always improve our writing with more revisions, which is what we have attempted to accomplish in this revision and will in the future revisions.

Results:

- Needs analysis by ecosystem or other global factors. Calculations of how much carbon this accounts for would also be really interesting and could add a new dimension to the study. Or at least better explain if/how the loss of biodiversity is meaningful to an ecosystem.

R: As advised, we have revised results accordingly (L145–151) and improved the description of our analysis (L288–305).

- A map of locations would be helpful

R: Great, we have moved the study site map from SI to Fig. 1.

- Throughout it needs to be made clear that this is potential function.

R: We are afraid that this comment is unclear to us.

- By experiment type – need to show that they responded the same

R: We have now added both the analysis with experimental system as the only predictor (Fig. S1) and carefully rewritten analysis section and results (L145–151, L288–305)

Reviewer #3 (Remarks to the Author):

I read the manuscript by Chen et al. on how loss of plant diversity detrimentally affects soil microbial biomass and some other soil microbial parameters. In general, I liked the manuscript and the efforts of the authors to compile such a database is praiseworthy. I, however, also have a number of issues with the presentation of the results and thus the overall message of the paper. I have detailed those below.

The main results (Fig. 1 and 2) presented by the authors are difficult to follow. First, how results in Fig. 1 are different compared to Fig. 2 (left panel) except that the units of effect size are different.

R: Both Figs. 1 and 2 resulted from one analysis since we tested the effects of predictors at the same time (see revised Eqn. 3). We have carefully revised the captions of both Figs. 2 and 3 (originally Figs. 1 and 2).

Second, if authors have information on species richness differences among studies, why not use those differences as a covariate/moderator to explain the variation in effect size which is shown in Fig. 1.

R: We presented the effect of the species richness in mixture in Fig. 2a. We have improved figure caption.

Third, I am unsure whether the stand age applies to all studies given some studies are from grasslands and/or greenhouse experiment. Thus, using this as a proxy of time or duration of the experiment is confusing and needs to be clarified.

R: For experiments in grasslands, croplands, and pots, stand age is same as experimental duration. For some studies conducted in forests, in which experiment began after stand establishment (although species mixtures vs. monocultures started at stand establishment), we think it is more appropriate to use stand age (L263–265).

And, finally clarification on stand age will help understand Fig. 3 and the results shown in Table S1, such as the significant interaction between stand age and species richness on two microbial parameters.

R: We have now improved all texts related to Fig. 4 in this submission (L126–129).

I appreciate the predictive approach used by the authors (Fig. 4), but this is problematic in the sense that environmental factors could not be included in this. Authors have themselves shown that certain environmental factors explain the variations in plant diversity effects on microbial parameters (Fig. S1, Table S4). Could we inform these statistical models with environmental parameters? At least, this needs to be discussed in the result and the discussion.

R: We have moved the text for the simulated responses to plant diversity loss (L152–159) after confirming the consistent responses of soil microbial biomass and respiration across all experimental systems (L145–147).

The writing needs to be improved for clear transition. For instance, the introduction section lacks providing underlying reasons for why we should expect plant diversity effects on soil microbial parameters. I have provided some specific examples in my other comments below.

R: We have worked thoroughly in this revision and thank you for specific comments below.

Other comments

Line 14: What does paired observation mean? Does it mean one monoculture vs. one mixture? I am not sure. Please clarify.

R: Paired observations are necessary for conducting meta-analysis to compare treatment effects. We have revised the abstract (L14) and throughout the entire manuscript to improve clarity, particularly about how we calculated lnRR (L272–279).

Line 18: gram-positive to gram-negative bacterial ratio.

R: Corrected.

Line 20: “the extent of biodiversity loss” is quite vague. Please clarify.

R: Removed.

Line 22: in ecosystems from colder climates.

R: Removed.

Line 34: by providing

R: Corrected.

Line 49: The abbreviation G+: G- ratio should be G+: G- bacteria ratio.

R: Corrected throughout.

Line 59: Metabolic quotient comes without a proper transition. Either provide some background before the term is introduced or begin with a new paragraph here.

R: Revised (L57–59).

Line 63: microbial C:N ratio.

R: Corrected.

Lines 68-73: I could not get the underlying reason why there is time lag in plant diversity effect on microbial biomass. Please clarify.

R: Revised (L65–66).

Lines 80-82: Unclear. Why under environmental stress, plant diversity would have higher positive interspecific plant interactions than plant monocultures?

R: Sorry for the ambiguous writing. Revised (L77–82).

Line 90: Here, we conducted...

R: Corrected.

Line 97: I don't think authors have developed an adequate background in the introduction for "the extent of diversity loss" effects on soil microbial communities. This needs to be properly developed for readers to understand what authors mean by the extent of biodiversity loss, why this could be important for soil microbial communities and how their study aims to tackle this.

R: Sorry for the ambiguous writing. We have revised this sentence (L97–98) with a proper introduction earlier (L72–82).

Line 101: Which microbial biomass is this..C or N?

R: Changed to "total soil microbial biomass" (L104).

Line 108-109: Why is this result relevant? And why not also run correlation for other response variables?

R: Removed.

References:

1.

Cardinale, B.J., Duffy, J.E., Gonzalez, A., Hooper, D.U., Perrings, C., Venail, P. *et al.* (2012). Biodiversity loss and its impact on humanity. *Nature*, 486, 59-67.

2.

Kardol, P., Fanin, N. & Wardle, D.A. (2018). Long-term effects of species loss on community properties across contrasting ecosystems. *Nature*, 557, 710-713.

Reviewers' Comments:

Reviewer #1:

Remarks to the Author:

The authors have provided a substantial revision, especially concerning the materials & methods section. The information is now clear and understandable. Added figures and tables greatly improve clarity.

However, some major issues remain and should be solved before publication in Nature communications. Especially some erroneous/incomplete statements are present in the introduction and these are reflected in the hypotheses and conclusions. One aspect in the data analyses and presentation of results remains unclear to me (Fig 2-Fig 3). Also, the discussion is rather descriptive lacking some mechanistical explanations of the results observed. These concerns are detailed below.

1. "Positive effects": As mentioned in for the first revision 'positive' or 'negative' effect can be ambiguous (positive=good, negative= bad). This was only partially revised, terms remain (i.e. L107, 126), especially in the title.
2. Title: In addition to the terms 'positive', the term 'communities' is misleading. see 3.
3. Microbial composition: this term generally refers to microbial community composition, with detailed taxonomic (molecular) analyses. While this term seems OK in the abstract (as the authors specify the term in the following sentence), I believe it is misleading in the general text and the exact terms fungi:bacteria ratio and G+/G- should be used.
4. L 16-17: I am not sure that "enhanced with" is a correct expression, and "microbial biomass increased in response to plant mixture" and "microbial biomass enhanced with species richness" is the same (see 5)
5. Fig. 2-3 and L 16-17: "We found that microbial biomass, bacterial biomass, fungal biomass, fungi:bacteria ratio and microbial respiration increased... in response to plant mixture" "The response of microbial biomass and respiration to plant mixture enhanced with the species richness in mixtures and over time." L96-97: "the effect of plant mixture would increase with the species richness in mixtures and over time": it is not clear what is meant by 'the effect would increase', does this imply a non-linear relationship between the response variable and species richness?
I am confused about these statements, as the difference between Fig 2 and 3 seems to me only that species richness and age are analysed separately for Fig. 3. However Fig 2 states 'The average effects of plant mixture on soil microbial attributes across all studies' (and comments address explicitly the parameters ie microbial biomass increased) and Fig 3 'The responses (slope coefficients) of microbial attributes to plant mixture' (and comments refer to effect size) of microbial biomass. But both graphs seem to represent the effect size (lnRR). This should be clarified.
6. Biodiversity (plant species) loss (Lines 21-22): as mentioned in the first revision, the term 'loss' is misleading, but this issue has not been addressed. The last sentence in the abstract still refers explicitly to species loss. To infer on species loss one would have to study an ecosystem and remove species, which is not the case here. This should be rewritten to refer to the finding that LOWER PLANT SPECIES DIVERSITY IS RELATED TO lower microbial biomass and respiration. Loss is only investigated in removal studies.

L325-335: remove loss -> (suggested changes in capital letters, text to remove in italics)
"To illustrate the effects of (OR: TO ILLUSTRATE THE RELATION BETWEEN) plant diversity *loss* on microbial biomass and respiration over time, we compared the lnRR when the plant richness in mixtures was R1 (*no species richness loss* ALL SPECIES PRESENT) and Ra (α % LOWER species richness loss). We assumed that the mean value of monocultures, Xc, did not vary with the number of monocultures of different species, which led to the following equation:

(4)

where P_a is the proportion of *remaining* microbial biomass or respiration under $\alpha\%$ LOWER plant species richness *loss* for a period of T , and other model terms were described in Eqn. 3. The detailed derivation process for Eqn. 4 is presented in Appendix 2. Based on Eqn. 4, we fitted curves for the decrease in microbial biomass and respiration over time when there was a 10%, 20%, 40%, and 80% DECREASE IN plant species richness *loss*."

And edit text in results, discussion and figure legends accordingly

7. "experimental systems": the authors misunderstood my previous comment, which recommended to use the same term throughout the manuscript, but I did not recommend this one. "Ecosystem type" is more appropriate as this study is neither experimental nor compares different systems of experiments. I therefore highly recommend changing this to 'ecosystem type'. In L88 one question of the authors is whether soil microbial biomass varies with 'experimental system', but the term 'experimental system' has only been explained by the authors as experimental grasslands vs containers (L72-75). In the discussion the meaning refers to different ecosystems and not experimental systems.

8. Do not see the link between issues addressed in L72-77 (experimental system, planted grasslands vs containers) and 77-82 (diversity effects in forests)

9. My concern was that the different studies (observation, experiments but where mixing was performed differently) have different meanings for the relationship, dependent on number of species in mixtures, the type of richness gradient (see Nadrowski 2010), but basic issues in the use of different setups have not been addressed (complementarity, facilitation, sampling effect, comprehensiveness, representativeness and orthogonality)

10. L52-54: "As different plant species have spatial partitioning in N uptake²⁶, plant communities of high diversity may have stronger competition for N over microorganisms, which thus increases microbial C:N ratio." "Therefore, our second hypothesis is that plant diversity would increase microbial C:N ratio and respiration, and decrease metabolic quotient."

This statement is wrong, N availability does not affect the microbial C:N ratio, which is mainly determined by physiological limits. The outcome of competition between plants and microorganisms will determine microbial biomass and net N mineralization rates, but only influence microbial C:N ratios to a limited extent. Variations in the C/N ratio are commonly related to shifts in microbial community composition (bacteria vs. fungi), since the fungi have higher carbon: element ratios (C:N: 5-17; N:P=15) than the bacteria (C:N: 6.5 N:P =7)

This point should be revised.

11. L42-43: "Further, as plant diversity is more directly linked to fungi than bacteria²²,": I believe that this is an over-generalization, the reference cited is for experimental grasslands and this finding cannot be extrapolated to all ecosystems. This should be edited (either change or provide more evidence for the statement). The statement also ignores that the fungi:bacteria ratio does depend on other factors such as soil pH, which can be influenced by individual species but may not be linked to 'diversity'.

12. L152-159: "10% decrease in plant species richness (from 100% to 90%) over one year": as mentioned in the first revision I do not believe that the data allows an extrapolation to predictions on what happens when species are lost over a certain amount of time. Data here are from different ecosystems at different ages (but not chronosequences), the component 'evolution with time' is not present and I am not convinced that it can be deduced from the data.

Species loss -> lower species richness, but time analysis is not valid

13. L177 facilitation between bacteria and fungi is not the only possible explanation, they might just both respond to more available substrate in mixtures due to higher productivity and litter

14. L322 "For the ease of interpretation, $\ln RR$ and its corresponding 95% confidence intervals (CIs) were transformed to a percentage change between monocultures and mixtures". I am not sure I understand this, is it really the change between mixtures and monocultures. $\ln RR$ is calculated from a theoretical expected value of the weighed values of monoculture species in the mixtures (X_c ; equation (1)). So this percentage represents the percentage change from the expected value in the mixture

based on monoculture values, thus assessingoveryielding? If this is the case, the concept of overyielding should be introduced.

15. Discussion: see comment G4 in first review. The authors respond to my comment on the weakness of the discussion by a clarification of the meaning of the response variable, but discussion was not improved.

16. L23 where \diamond and (where refers to a location which is not the case here)

17. Containers (L 21) \diamond planted containers

18. Plant mixture \diamond plant mixtures

19. L45: (G+) bacteria is \diamond (G+) bacteria are

20. L47: reduced moisture associated with plant diversity loss \diamond reduced moisture associated with lower plant diversity

21. L307 verb is missing

22. L149-151: "For microbial respiration, the response of microbial respiration to plant mixture increased more strongly with the species richness in mixtures with decreasing mean annual temperature": this is really difficult to understand, please revise the sentence

23. L167: what is a global ecosystem? remove global here speak about ecosystems, as before, it would be better to use ecosystem type vs experimental type

24. L195: more pronounced richness effects over time \diamond more pronounced richness effects in older stands

25. L197-199 and L203-206: repetition

Reviewer #2:

Remarks to the Author:

Overall the manuscript is much improved, and again the data and results are impressive and of interest to NatComm. My concern is how the results are presented. After the rewrite, the manuscript is more clear, but another round of edits is needed.

First, from the title, abstract and introduction the text suggests that composition is examined. At a time where high-throughput sequencing is commonplace, the use of the word 'composition' is misleading in the context of G+/- . The main results are really about biomass and respiration. The introduction text should reflect this and not make broad statements about composition/abundances.

Hypotheses – the intro is much improved, yet again there is confusion about the hypothesis in the earlier paragraphs (which really are predictions). I have added suggestions below on how this can be addressed.

Finally, the main questions and why this mesocosm is important at a large scale remains ambiguous. Indeed, the dataset is impressive, and yes, there are global patterns. But WHY does this matter. There is little comment in the intro and none in the discussion. This would greatly improve the manuscript.

More specific comments below:

Title "globally positive effects" does not make sense. Maybe try:

- Globally, plant diversity has positive influence on soil microbe communities
- Leave our 'positive' and just say "Global effects of.."

Abstract

L19 – rewrite, unclear.

Introduction.

- Sentence 2 does not relate to either sentence 1 or 3

L31- this is a huge overstatement, we hardly know how plants species support specific abundances/composition.

L34 divergent is the wrong word because responses are not binary, they are quite variable...

L35-36 this is all about biomass not composition/abundance which is misleading from the previous sentence on line 31

L38: Add 'For example' before plant productivity

L48: These are predictions for a hypothesis stating 'If plant biodiversity is important for microorganisms, than...' Expand on this to get at the mechanisms. The same is true for the second 'hypothesis'.

L70-71: Again, this is a prediction.

L87-89: Until here the major questions have remained unclear. This needs to be moved up to the beginning, so the background is put into better context. (It would fit well after L36)

L93-98: these are much better. Therefore, in the previous section where the word 'hypothesis' is used, consider changing the sentence to something along the lines of "therefore with increased plant diversity we would expect"

L156: Consider rewording

L200: I think it is meant to say 'lack of effects in certain studies'? Or studies that do not observe an effect of diversity on biomass...

L208: was to were

L216-224: this is only restating the results above. A concluding paragraph relating these results to a larger picture – global carbon/global biodiversity would be much more compelling.

Reviewer #3:

Remarks to the Author:

The manuscript has definitely improved from the previous version. I, however, still find presenting effect sizes in different units not the best approach. For instance, why not use % change as shown in figure 2 also for figure 3 and figure 4 instead of log response ratio. The other issue that I don't think is resolved is treating time effect (e.g. figure 4) differently for forests and grasslands. We need to see whether time effects of plant diversity on microbial biomass differ between grasslands and forests. This also relates to the reviewer 1 remark about the ecosystem type. The time effect should be tested separately for forests and grasslands.

Reviewer #1 (Remarks to the Author):

The authors have provided a substantial revision, especially concerning the materials & methods section. The information is now clear and understandable. Added figures and tables greatly improve clarity.

However, some major issues remain and should be solved before publication in Nature communications. Especially some erroneous/incomplete statements are present in the introduction and these are reflected in the hypotheses and conclusions. One aspect in the data analyses and presentation of results remains unclear to me (Fig 2-Fig 3). Also, the discussion is rather descriptive lacking some mechanistical explanations of the results observed. These concerns are detailed below.

1. “Positive effects”: As mentioned in for the first revision ‘positive’ or ‘negative’ effect can be ambiguous (positive=good, negative= bad). This was only partially revised, terms remain (i.e. L107, 126), especially in the title.

R: ‘Positive’ and ‘negative’ effects have been revised throughout.

2. Title: In addition to the terms ‘positive’, the term ‘communities’ is misleading. see 3.

R: “Positive’ has been removed.

3. Microbial composition: this term generally refers to microbial community composition, with detailed taxonomic (molecular) analyses. While this term seems OK in the abstract (as the authors specify the term in the following sentence), I believe it is misleading in the general text and the exact terms fungi:bacteria ratio and G+/G- should be used.

R: As recommended, we have now used the exact terms.

4. L 16-17: I am not sure that “enhanced with” is a correct expression, and “microbial biomass increased in response to plant mixture” and “microbial biomass enhanced with species richness” is the same (see 5)

R: Revised to improve clarity (L17-18)

5. Fig. 2-3 and L 16-17: “We found that microbial biomass, bacterial biomass, fungal biomass, fungi:bacteria ratio and microbial respiration increased... in response to plant mixture” “The response of microbial biomass and respiration to plant mixture enhanced with the species richness in mixtures and over time.” L96-97: “the effect of plant mixture would increase with the species richness in mixtures and over time”: it is not clear what is meant by ‘the effect would increase’, does this imply a non-linear relationship between the response variable and species richness?

R: To improve clarity, we have reworded all mentioned text (L17-18, L126-128). In our analysis, we simultaneously determined the average effect of plant mixtures (beta0) (presented in Fig. 2), the effects associated with the variation in the species richness in mixtures (beta1) (note that the richness varied from 2 to 60 species) (presented in Fig. 3a), and the effects associated with stand age (beta2) (note that stand age varied from 1 to 130 years) (presented in Fig. 3b) (see eqn. 3 in Methods). We have revised the figure captions to improve readability.

I am confused about these statements, as the difference between Fig 2 and 3 seems to me only that species richness and age are analysed separately for Fig. 3. However Fig 2 states ‘The average effects of plant mixture on soil microbial attributes across all studies’ (and comments address explicitly the parameters ie microbial biomass increased) and Fig 3 ‘The responses (slope coefficients) of microbial attributes to plant mixture’ (and comments refer to effect size) of microbial biomass. But both graphs seem to represent the effect size (lnRR). This should be clarified.

R: Sorry for the lack of clarity. In this revision, we have revised figure captions to improve clarity (Figs. 1-4).

6. Biodiversity (plant species) loss (Lines 21-22): as mentioned in the first revision, the term 'loss' is misleading, but this issue has not been addressed. The last sentence in the abstract still refers explicitly to species loss. To infer on species loss one would have to study an ecosystem and remove species, which is not the case here. This should be rewritten to refer to the finding that LOWER PLANT SPECIES DIVERSITY IS RELATED TO lower microbial biomass and respiration. Loss is only investigated in removal studies.

R: Thank you for your advice. We have checked species loss term throughout and made changes as recommended.

L325-335: remove loss -> (suggested changes in capital letters, text to remove in italics)

“To illustrate the effects of (OR: TO ILLUSTRATE THE RELATION BETWEEN) plant diversity loss on microbial biomass and respiration over time, we compared the $\ln RR$ when the plant richness in mixtures was R_1 (no species richness loss ALL SPECIES PRESENT) and R_α ($\alpha\%$ LOWER species richness loss). We assumed that the mean value of monocultures, X_c , did not vary with the number of monocultures of different species, which led to the following equation:

(4)

where P_α is the proportion of remaining microbial biomass or respiration under $\alpha\%$ LOWER plant species richness loss for a period of T , and other model terms were described in Eqn. 3. The detailed derivation process for Eqn. 4 is presented in Appendix 2. Based on Eqn. 4, we fitted curves for the decrease in microbial biomass and respiration over time when there was a 10%, 20%, 40%, and 80% DECREASE IN plant species richness loss.”

And edit text in results, discussion and figure legends accordingly

R: As recommended, we revised the text throughout.

7. “experimental systems”: the authors misunderstood my previous comment, which recommended to use the same term throughout the manuscript, but I did not recommend this one. “Ecosystem type” is more appropriate as this study is neither experimental nor compares different systems of experiments. I therefore highly recommend changing this to ‘ecosystem type’. In L88 one question of the authors is whether soil microbial biomass varies with ‘experimental system’, but the term ‘experimental system’ has only been explained by the authors as experimental grasslands vs containers (L72-75). In the discussion the meaning refers to different ecosystems and not experimental systems.

R: Sorry for our misunderstanding in our previous revision. In this revision, we have changed experimental systems back to ecosystem types throughout.

8. Do not see the link between issues addressed in L72-77 (experimental system, planted grasslands vs containers) and 77-82 (diversity effects in forests)

R: We have reworded the entire paragraph to improve flow (L72-82).

9. My concern was that the different studies (observation, experiments but where mixing was performed differently) have different meanings for the relationship, dependent on number of species in mixtures, the type of richness gradient (see Nadrowski 2010), but basic issues in the use of different setups have not been addressed (complementarity, facilitation, sampling effect, comprehensiveness, representativeness and orthogonality)

R: Thank you for raising two important issues here: 1) observational versus experimental studies and 2) the partitioning of diversity effects. For 1), we have clarified our text (L72-82) and our

analysis explicitly tested the ideas (L151-158). For 2), we acknowledge strong influences of species identity (or composition), i.e., sampling/selection effect on ecosystem function. For this reason, Loreau and Hector (2001) developed the method to partition selection and complementarity. We used the method by Loreau and Hector (2001) to quantify complementarity effect since our InRR has accounted for the selection effect. To further clarify, we have added a paragraph in Discussion (L220-233) and one in Data analysis (L296-297).

10. L52-54: "As different plant species have spatial partitioning in N uptake²⁶, plant communities of high diversity may have stronger competition for N over microorganisms, which thus increases microbial C:N ratio." "Therefore, our second hypothesis is that plant diversity would increase microbial C:N ratio and respiration, and decrease metabolic quotient."

This statement is wrong, N availability does not affect the microbial C:N ratio, which is mainly determined by physiological limits. The outcome of competition between plants and microorganisms will determine microbial biomass and net N mineralization rates, but only influence microbial C:N ratios to a limited extent. Variations in the C/N ratio are commonly related to shifts in microbial community composition (bacteria vs. fungi), since the fungi have higher carbon: element ratios (C:N: 5-17; N:P=15) than the bacteria (C:N: 6.5 N:P =7)

This point should be revised.

R: Thank you for your advice. We have revised the relevant text (L48-50) based on your recommendation.

11. L42-43: "Further, as plant diversity is more directly linked to fungi than bacteria²²,": I believe that this is an over-generalization, the reference cited is for experimental grasslands and this finding cannot be extrapolated to all ecosystems. This should be edited (either change or provide more evidence for the statement). The statement also ignores that the fungi:bacteria ratio does depend on other factors such as soil pH, which can be influenced by individual species but may not be linked to 'diversity'.

R: We have revised the sentence and changed the cited reference (Chung *et al.* 2007), which shows experimental results of species richness on fungal and bacterial abundance. To not overgeneralize, we presented this sentence as a hypothesis (L38-40). Indeed, fungi:bacteria ratio would be affected by soil pH and overstory composition (i.e., species identity effect). Our analysis selected all original studies, each compared mixtures versus monocultures on the same climate and soil conditions including soil pH, and we have removed the effect of species identity in our calculation of InRR (see Data analysis).

12. L152-159: "10% decrease in plant species richness (from 100% to 90%) over one year": as mentioned in the first revision I do not believe that the data allows an extrapolation to predictions on what happens when species are lost over a certain amount of time. Data here are from different ecosystems at different ages (but not chronosequences), the component 'evolution with time' is not present and I am not convinced that it can be deduced from the data.

Species loss -> lower species richness, but time analysis is not valid

R: As recommended, we have changed species loss to lower species richness throughout the entire text.

Concerning time analysis, following the recommendation of textbooks (Hicks & Turner 1999; Cohen *et al.* 2013; Schmidt & Hunter 2014), we used mixed effect models with the restricted maximum likelihood estimation and "study" as the random effect to account for the lack of independence among observations within each study. The random effect standardizes observations to allow determination of fixed effects including time (see Eqn. 3). We further bootstrapped our coefficient estimates to ensure the best coefficient estimates.

13. L177 facilitation between bacteria and fungi is not the only possible explanation, they might just both respond to more available substrate in mixtures due to higher productivity and litter

R: We agree. This explanation has been revised as recommended (L181-185).

14. L322 "For the ease of interpretation, LnRR and its corresponding 95% confidence intervals (CIs) were transformed to a percentage change between monocultures and mixtures". I am not sure I understand this, is it really the change between mixtures and monocultures. LnRR is calculated from a theoretical expected value of the weighed values of monoculture species in the mixtures (X_c ; equation (1)). So this percentage represents the percentage change from the expected value in the mixture based on monoculture values, thus assessing overyielding? If this is the case, the concept of overyielding should be introduced.

R: Great point, added as recommended (L343).

15. Discussion: see comment G4 in first review. The authors respond to my comment on the weakness of the discussion by a clarification of the meaning of the response variable, but discussion was not improved.

R: Sorry for our ignorance. In this revision, we reflected more about your original comment by adding a paragraph in Discussion (L220-233). We understand the importance of functional diversity. However, we do not have such data. We have been explicit in all contexts that we deal with species diversity.

16. L23 where -> and (where refers to a location which is not the case here)

R: Corrected.

17. Containers (L 21) -> planted containers

R: Corrected throughout the manuscript.

18. Plant mixture -> plant mixtures

R: Corrected throughout the manuscript and figures.

19. L45: (G+) bacteria is -> (G+) bacteria are

R: Corrected.

20. L47: reduced moisture associated with plant diversity loss -> reduced moisture associated with lower plant diversity

R: Corrected.

21. L307 verb is missing

R: Corrected.

22. L149-151: "For microbial respiration, the response of microbial respiration to plant mixture increased more strongly with the species richness in mixtures with decreasing mean annual temperature": this is really difficult to understand, please revise the sentence

R: Revised (L156-158).

23. L167: what is a global ecosystem? remove global here speak about ecosystems, as before, it would be better to use ecosystem type vs experimental type

R: 'Global' has been removed. 'Experimental type' has been changed to 'ecosystem type' throughout the manuscript.

24. L195: more pronounced richness effects over time more pronounced richness effects in older stands

R: Revised as recommended (L198-199).

25. L197-199 and L203-206: repetition

R: The sentence in original L203-206 has been move up to replace the original L197-199 (L201-204).

Reviewer #2 (Remarks to the Author):

Overall the manuscript is much improved, and again the data and results are impressive and of interest to NatComm. My concern is how the results are presented. After the rewrite, the manuscript is more clear, but another round of edits is needed.

R: Thank you.

First, from the title, abstract and introduction the text suggests that composition is examined. At a time where high-throughput sequencing is commonplace, the use of the word 'composition' is misleading in the context of G+/- . The main results are really about biomass and respiration. The introduction text should reflect this and not make broad statements about composition/abundances.

R: During this revision, we have narrowed down to soil microbial attributes we have data for.

Hypotheses – the intro is much improved, yet again there is confusion about the hypothesis in the earlier paragraphs (which really are predictions). I have added suggestions below on how this can be addressed.

R: Thank you for advice.

Finally, the main questions and why this mesocosm is important at a large scale remains ambiguous. Indeed, the dataset is impressive, and yes, there are global patterns. But WHY does this matter. There is little comment in the intro and none in the discussion. This would greatly improve the manuscript.

R: Great advice. We have now added implications of our results (L240-244). We also highlight the importance in Abstract (L11-12) and Introduction (L25-26).

More specific comments below:

Title "globally positive effects" does not make sense. Maybe try:

- Globally, plant diversity has positive influence on soil microbe communities
- Leave our 'positive' and just say "Global effects of.."

R: Changed as recommended. Thank you.

Abstract

L19 – rewrite, unclear.

R: Revised (L17-18)

Introduction.

- Sentence 2 does not relate to either sentence 1 or 3

R: This sentence has been removed for better logic flow.

L31- this is a huge overstatement, we hardly know how plants species support specific abundances/composition.

R: We have deleted this sentence and incorporate ideas in the next paragraph.

L34 divergent is the wrong word because responses are not binary, they are quite variable...

R: Deleted.

L35-36 this is all about biomass not composition/abundance which is misleading from the previous sentence on line 31

R: Deleted.

L38: Add 'For example' before plant productivity

R: The sentence has been deleted .

L48: These are predictions for a hypothesis stating 'If plant biodiversity is important for microorganisms, than...' Expand on this to get at the mechanisms. The same is true for the second 'hypothesis'.

R: Revised as recommended (L44-57, L57-60).

L70-71: Again, this is a prediction.

R: Revised (L69-71).

L87-89: Until here the major questions have remained unclear. This needs to be moved up to the beginning, so the background is put into better context. (It would fit well after L36)

R: We have reworked the first paragraph closing with a statement in L29-32.

L93-98: these are much better. Therefore, in the previous section where the word 'hypothesis' is used, consider changing the sentence to something along the lines of "therefore with increased plant diversity we would expect"

R: Revised as recommended.

L156: Consider rewording

R: Deleted.

L200: I think it is meant to say 'lack of effects in certain studies'? Or studies that do not observe an effect of diversity on biomass...

R: 'In certain studies' was added here (L205).

L208: was to were

R: Corrected.

L216-224: this is only restating the results above. A concluding paragraph relating these results to a larger picture – global carbon/global biodiversity would be much more compelling.

R: Great advice. We have now added text to make our results more compelling (L240-244).

Reviewer #3 (Remarks to the Author):

The manuscript has definitely improved from the previous version. I, however, still find presenting effect sizes in different units not the best approach. For instance, why not use % change as shown in figure 2 also for figure 3 and figure 4 instead of log response ratio.

R: Changed as recommended (Figs 3 and 4).

The other issue that I don't think is resolved is treating time effect (e.g. figure 4) differently for forests and grasslands. We need to see whether time effects of plant diversity on microbial biomass differ between grasslands and forests. This also relates to the reviewer 1 remark about the ecosystem type. The time effect should be tested separately for forests and grasslands.

R: In our previous submission, we indeed tested whether the time effect differed among ecosystem types, i.e., $\beta_5 \cdot A \times E$ and $\beta_6 \cdot R \times A \times E$ in eqn. 3 (see Methods). Further to your comment here, we separately tested the time effect between forests and grasslands. In both ecosystem types, effect size increases with stand age/time, but the response slopes do not differ significantly ($P = 0.588$), and we show this graphically below.

References

- Chung, H.G., Zak, D.R., Reich, P.B. & Ellsworth, D.S. (2007) Plant species richness, elevated CO₂, and atmospheric nitrogen deposition alter soil microbial community composition and function. *Global Change Biology*, **13**, 980-989.
- Cohen, J., Cohen, P., West, S.G. & Aiken, L.S. (2013) *Applied Multiple Regression/Correlation Analysis for the Behavioral Sciences*. Routledge.

- Hicks, C.R. & Turner, K.V. (1999) *Fundamental Concepts in the Design of Experiments*, 5th edn. Oxford University Press, New York, New York.
- Loreau, M. & Hector, A. (2001) Partitioning selection and complementarity in biodiversity experiments. *Nature*, **412**, 72-76.
- Schmidt, F.L. & Hunter, J.E. (2014) *Methods of meta-analysis: Correcting error and bias in research findings*. Sage publications.

Reviewers' Comments:

Reviewer #1:

Remarks to the Author:

This second review clarified most issues. A few comments were incompletely addressed and they are listed below:

1. As mentioned in previous revisions and also by reviewer #2, terms used are misleading with regard to the content/topic of the paper: 'microbial communities' (title, L218), 'relative abundance of microbial taxa' (L22, L34, L89,L248), 'composition' (L31) generally refer to microbial community composition, with detailed taxonomic (molecular) analyses. The exact terms fungi:bacteria ratio and G+/G- should be used. Although authors claim in the rebuttal letter that this has been corrected throughout, it is not the case.
2. Microbial C:N ratio:this issue was partially revised (L49-50, L57-60).However the sentence L50-51 still suggests that higher N availability would change the microbial C:N ratio, which is incorrect ("plant diversity may have little effect on microbial C:N because of increased N availability for soil microorganisms").
3. L23, L70, L97, L236, Figure 5, L244 'over time' \diamond 'stand age'. The authors mention in the revision that the mixed model allows analysis for time, which is correct. However, the parameter 'time' does not reflect the evolution over time (chronosequence) but different stand ages.
4. L11 'form the foundation of biogeochemical cycling': strange expression (in particular 'form')
 \diamond rephrase
5. L21-23 this paragraph of the summary repeats the previous one (L15-20).It would be more meaningful to end the summary with a statement on the broad significance of the results.

REVIEWERS' COMMENTS: Reviewer #1 (Remarks to the Author):

This second review clarified most issues. A few comments were incompletely addressed and they are listed below:

1. As mentioned in previous revisions and also by reviewer #2, terms used are misleading with regard to the content/topic of the paper: 'microbial communities' (title, L218), 'relative abundance of microbial taxa' (L22, L34, L89, L248), 'composition' (L31) generally refer to microbial community composition, with detailed taxonomic (molecular) analyses. The exact terms fungi:bacteria ratio and G+/G- should be used. Although authors claim in the rebuttal letter that this has been corrected throughout, it is not the case.

R: All these terms have been replaced with fungi:bacteria ratio and G+:G- bacteria ratio in these sentences (Line 229, Line33-34, Line 86, Line 259, Line 31). The title and sentence in Line 22 have been rephrased.

2. Microbial C:N ratio: this issue was partially revised (L49-50, L57-60). However the sentence L50-51 still suggests that higher N availability would change the microbial C:N ratio, which is incorrect ("plant diversity may have little effect on microbial C:N because of increased N availability for soil microorganisms").

R: This sentence has been removed as recommended.

3. L23, L70, L97, L236, Figure 5, L244 'over time' \diamond 'stand age'. The authors mention in the revision that the mixed model allows analysis for time, which is correct. However, the parameter 'time' does not reflect the evolution over time (chronosequence) but different stand ages.

R: 'over time' has been changed to 'with stand age' in these sentences as recommended (Line 67, 94, 247, Figure 5 legend and 255). Sentence in Line 23 has been rephrased.

4. L11 'form the foundation of biogeochemical cycling': strange expression (in particular 'form') \diamond rephrase

R: This sentence has been rephrased as recommended (Line 12).

5. L21-23 this paragraph of the summary repeats the previous one (L15-20). It would be more meaningful to end the summary with a statement on the broad significance of the results.

R: We have made a statement on the broader significance of the results as recommended (Line 21-23).